# Measurement of CO$_2$ Emissions by the Operation of Freight Transport in Mexican Road Corridors

Juan F. Mendoza-Sanchez [1,*] , Elia M. Alonso-Guzman [2,*] , Wilfrido Martinez-Molina [2] , Hugo L. Chavez-Garcia [2] , Rafael Soto-Espitia [2], Saúl A. Obregón-Biosca [3] and Horacio Delgado-Alamilla [1]

1   Mexican Institute of Transportation, Queretaro 76703, Mexico; horacio.delgado@imt.mx
2   Faculty of Civil Engineering, Michoacan University of San Nicolas of Hidalgo, Morelia 58030, Mexico;
    wilfrido.martinez@umich.mx (W.M.-M.); luis.chavez@umich.mx (H.L.C.-G.); rsoto@umich.mx (R.S.-E.)
3   Faculty of Engineering, Autonomous University of Queretaro, Queretaro 76010, Mexico
*   Correspondence: fernando.mendoza@imt.mx (J.F.M.-S.); elia.alonso@umich.mx (E.M.A.-G.)

**Abstract:** The freight transport industry in Mexico has grown significantly since the establishment of trade agreements in North America, which has brought significant environmental consequences to the main transport corridors. This paper proposes a methodology for the estimation of emissions for freight vehicles on road transportation corridors. The variables included in this analysis allow adequate characterization of the conditions of the vehicle fleet, the geometry and the quality of the road, the environment, and the average annual daily traffic (AADT) of heavy vehicles. The results were structured to show two indicators, the amount of CO$_2$ emissions per kilometer and the amount of emissions per tonne transported. These results will allow establishing a baseline of CO$_2$ emissions through which we can implement actions in the road transport sector to reduce greenhouse gases (GHG) to mitigate climate change and develop parameter values for use in Cost Benefit Analysis. The indicators can also be applied to geospatial modeling of emissions in road transport corridors and forecast its growth.

**Keywords:** freight road transport; CO$_2$ emissions; climate change mitigation

## 1. Introduction

The challenge in recent years for most countries on environmental issues is achieving global action against climate change. In order to take action, governments have developed a set of targets aimed at reducing emissions of greenhouse gases (GHG) in programs and strategies at the national level, which are documented in several research studies [1–3]. At the same time, governments are being prepared to implement adaptation measures to reduce the vulnerability of their critical systems, such as transport [4]. The World Road Association has documented some examples in countries such as England, Scotland, Japan, the USA, Australia, and Germany [5].

Several countries have found that the largest source of GHG generation is transportation. In this sense, the Organization for Economic Co-operation and Development (OECD) highlights that the sector accounted for approximately 13% of overall GHG emissions and 24% of CO$_2$ emissions from fossil fuel combustion in 2006 [6]. Considering the development path of the transport sector in developed countries, there is little doubt that the road transport sector emissions will be dominant [7]. Nevertheless, a major concern for road freight transport is the fact that energy consumption is growing at a faster rate than energy used by cars and buses [8]. Freight transportation consumes about 43% of all fuel used in transportation and is responsible for a major share of transport-related emissions of nitrogen oxides, unburned hydrocarbons, and fine particulate matter [9].

The freight transport industry in Mexico has grown significantly since the establishment of the North American Free Trade Agreement in 1994 (NAFTA, now USMCA), which has brought significant environmental consequences to the main transport corridors [10].

Commodity movements in North America are concentrated in trade between Canada and the USA, and between Mexico and the USA.

According to the Secretariat of Infrastructure, Communications and Transportation in Mexico (SICT, 2021), around 700 million tonnes of goods were carried by different modes of transport in 2020 (7% less than in 2019 due to the pandemic of COVID-19) [11]. More than 77% of goods are transported by heavy-duty vehicles, which is currently the source generating the highest number of emissions and other environmental impacts such as noise, accidents, solid waste generation [12,13], and climate change [14]. Trucks leading the movement in freight transportation (by value) among the three-country members of the North American Free Trade Agreement (NAFTA), now called the United States–Mexico–Canada Agreement (USMCA) [15]. Worldwide, it is highlighted that there is significant attention on road transport, as this is the dominant mode of freight movement [16,17]. In this sense, the Mexican transport sector is one of the largest sources of GHG contribution, which represents 25% of the national total, with 171 Mt $CO_2$eq issued in 2015, distributed by roads (159.4 Mt $CO_2$eq), maritime (2.6 Mt $CO_2$eq), railroad (2.5 Mt $CO_2$eq), electric (not significant), and air (6.3 Mt $CO_2$eq) with information of the Sixth National Communication of Mexico [18].

Most studies to measure the impact of climate change on the transport sector are based on the application of methodologies that have set the Intergovernmental Panel on Climate Change (IPCC) called National Emissions Inventories (developed by 185 countries that are members of Parties Conferences in the United Nations Framework Convention on Climate Change). The methodologies for the inventory of emissions from the transport sector have a top-down approach, converting the energy consumption of the country through emission factors in GHG emissions [19]. Using this approach, some studies have estimated emissions in the transport sector [20,21].

Researchers and technical agencies of various countries have developed inventories with more detailed information, using methodologies with a bottom-up approach [22–24]. This type of inventory in the transport sector focuses on ground transportation in cities, but it does not usually include intercity transportation (road transport).

In-depth studies of emission inventories on roads have been identified specifically for freight transport. A study based on surveys applied to transport companies was carried out, where information was gathered about energy consumption, distance traveled, and vehicle performance by Heavy Goods Vehicle (HGV) weight class. The methods of calculation in the UK use four approaches (described in Section 2) to estimate $CO_2$ emissions from road freight transport. The conclusion of this study describes the need to reconcile differences in truck-km estimates based on surveys and roadside traffic counts depending on the scope of the analysis [16]. Another study evaluates the environmental impact of a major freight corridor in California, particularly associated with heavy-duty diesel trucks. The baseline was developed using a traffic microsimulation tool (TransModeler). In this study, vehicle emissions depend on vehicle type, model year, and fuel type. This methodology estimates instantaneous emissions and the fuel consumption rate for different vehicle types and estimates emissions under various traffic operational scenarios (such as congestion, traffic signals, and High-occupancy vehicle lanes). The results show that utility is applied to this process to evaluate scenarios; however, they exclude infrastructure characteristics [25].

The Commission for Environmental Cooperation developed a study to determine the actual impact of $CO_2$ emissions in the transport corridors of North America and its future impact [10]. The methodology developed a bottom-up approach; however, a few variables such as traffic flow, emission rates, corridor data, and truck data were considered in the analysis in order to establish comparability between the three countries of the region (Canada, Mexico, and the United States of America). The information used in this study also does not include detailed information about road characteristics to obtain a reliable measure of emissions to construct a baseline.

To implement emission reduction strategies, it is necessary to have a reliable emission inventory that allows the establishment of a baseline in order to measure the impact of

the action on the mitigation of climate change. In order to reduce GHG emissions in freight transportation, it requires using appropriate emission models and integrating data into the bottom-up methodology in the planning process, to develop a reliable emission inventory [26].

Rapid urbanization and economic growth in developing countries have spurred air pollution, especially in the road transport sector. The increasing demand for petroleum-based fuels and their combustion in internal combustion (IC) engines have adverse effects on air quality, human health, and global warming [27]. Addressing climate change in developing countries poses a fundamentally different challenge. Developing countries will continue to increase their emissions due to economic growth and a better quality of life [28].

Forecasts of emission in Mexico trends shown in Figure 1 show a significant growth in GHG emissions [29]. Appropriate strategies are required to reduce these future increases, such as the ones analyzed [30], where 21 GHG mitigation measures were applied to reduce transport sector GHG emissions in Mexico. This, therefore, results in the need to develop a methodology to estimate emissions in transport corridors in developing countries such as Mexico through a bottom-up approach.

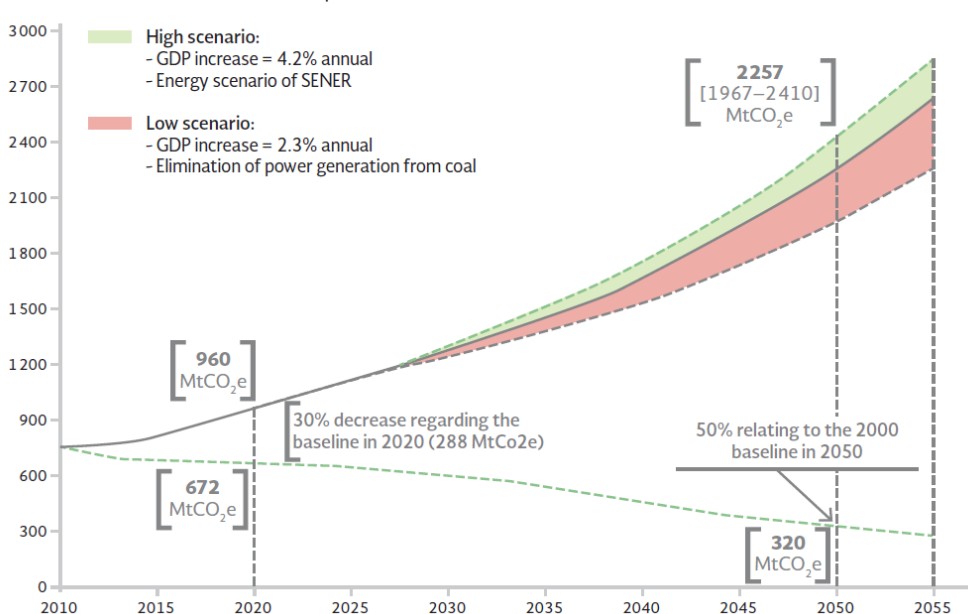

**Figure 1.** Baseline and objective trajectory of GHG emissions in Mexico 2010–2050. Source: SEMAR-NAT, 2013. Notes: Gross Domestic Product (GDP); Ministry of Energy (SENER).

The aim of this study, on one hand, is to develop a methodology with a bottom-up approach, with as many variables of road transportation, and to obtain an emission inventory for measuring $CO_2$ emissions of freight transport in road corridors for Mexico. Additionally, this study seeks to establish environmental indicators with the results applicable to $CO_2$ in transport corridors.

## 2. Review of Literature

It is noted that bottom-up methodologies are supported using models for estimating emissions, which are based on various factors that influence energy consumption. A review of several studies to compile various factors that influence energy consumption and emissions generation during freight transport was conducted. The factors were classified into five categories: vehicle (vehicle curb weight, vehicle shape, engine size/type, engine temperature, transmission, fuel type/composition, age, maintenance, and accessories); environment (roadway gradient, pavement type, ambient temperature, altitude, wind conditions, humidity, and surface conditions); traffic (speed, acceleration/deceleration, and congestion); driver (driver

aggressiveness, gear selection, and idle time) and operation (fleet size an mix, payload, empty kilometers, and number of stops) [31].

Under the MEET project "Methodologies for estimating air pollutant emissions from transport", emissions from road transport are calculated with the emission factor and the activity (distance traveled per time unit). Emission factors were estimated through driving cycles by the Transportation Research Laboratory in the UK, particularly for heavy-duty vehicles, and include variables such as road gradient, speed, weight vehicle, and empty vehicles [32].

Four approaches have been adopted in the UK to estimate $CO_2$ emissions from road freight transport. Approach one employs survey data from the Continuing Survey of Road Goods Transport (CSRGT), using the average efficiency and average annual distance traveled to obtain a measure of total fuel consumption. Approach two utilizes a combination of traffic flow estimates of truck-km with survey-based fuel efficiency estimates. Approach three is similar to approach two, where only the efficiency estimates are based on vehicle test cycles. Approach four estimates of total fuel sales to HGV, which are multiplied by a $CO_2$ conversion ratio similar to IPPC methodologies [16]. Emission inventories based on IPCC methodologies involve the multiplication of energy consumption by an emission factor according to the activity.

In Colombia, to estimate emissions by trucks, the Handbook Emission Factors for Road Transport (HBEFA) was used [33]. To accomplish this, information on each vehicle was collected, including model, year, type of fuel used, Euro standard, and engine power trucks. The commercial vehicles were equipped with a Global Positioning System (GPS) to obtain second-to-second speed, location, acceleration, and deceleration; the rest of the data were provided by the vehicles' owners.

It was identified that some emissions models are integrated into a computer program and include different factors to estimate emissions, described as the most relevant to the transport sector. MOBILE6 Vehicle Emission Modeling Software is an emission factor model for predicting gram-per-mile emissions developed by the Environmental Protection Agency [34]. The next emission model developed by EPA is the "Motor Vehicle Emission Simulator (MOVES)", which is a state-of-the-science emission modeling system that estimates emissions for mobile sources at the national, county, and project level for criteria air pollutants, GHG, and air toxins [35]. Both models have a useful application for estimating emissions from transportation in cities but are more complicated to use for estimating emissions on roads.

The Highway Development and Management System is a decision tool to help with road investment choices (HDM-4). One part of the model allows estimating the social and environmental effects of investment through the sub-model of energy and environmental effects that permit estimating some type of emissions such as $CO_2$ [36]. Another model used in Europe is the COPERT 4 (Computer Program to Calculate Emission from Road Traffic), which is software to estimate air pollutant emissions from road transport. The development of COPERT 4 has been financed by the European Environment Agency [37]. The COPERT 4 methodology has been developed for the compilation of national inventories or specific studies such as the one developed to assess the sector's contribution to the total national emissions budget (inventory) of the different fuels [38].

Using a greater number of variables allows a more reliable estimation of emissions, but this must be consistent with the country's ability to provide information. EPA models (MOBILE6 and MOVES) are widely used in Mexico cities; however, the information for roads is not easy to complete. While the COPERT 4 model can be used, emission factors are more commensurate with the European context. The HDM-4 model was made for roads, and variables for estimating emissions are available in Mexico and are detailed below.

## 3. Methodology and Data Collection

The framework for estimating emissions from freight transport is mainly defined by the infrastructure (transport corridors), traffic data (annual average daily traffic, vehicle age,

and percentage of vehicles loaded), light and heavy vehicles, geography and environmental characterization, and design infrastructure (road design and surface pavement conditions).

### 3.1. Transport Corridors

A corridor is a broad geographic area, defined by logical, existing, and forecasted travel patterns served by various modal transportation systems that provide important connections within and between regions of the State for people, goods, and services [39]. Travel within the corridor may include vehicle, rail, transit, water, air, or non-motorized means. In the same sense, a transport corridor is the area of origin and destination of traffic flows [40].

In Mexico, transport corridors are defined by the largest movement of freight and passengers across the road network. The first transport corridors were defined through origin–destination data and cargo transported [41]. Transport corridors are placed into a hierarchy using land use models, incorporating the economic value of the freight transported. The results can then be applied to the first strategic planning for the expansion of the corridors in the country. Nowadays, the network is composed of 16 large road corridors that connect the main ports and border crossings, in addition to providing accessibility to major cities that make up the country and Mexico City. Figure 2 shows the main transport corridors in Mexico.

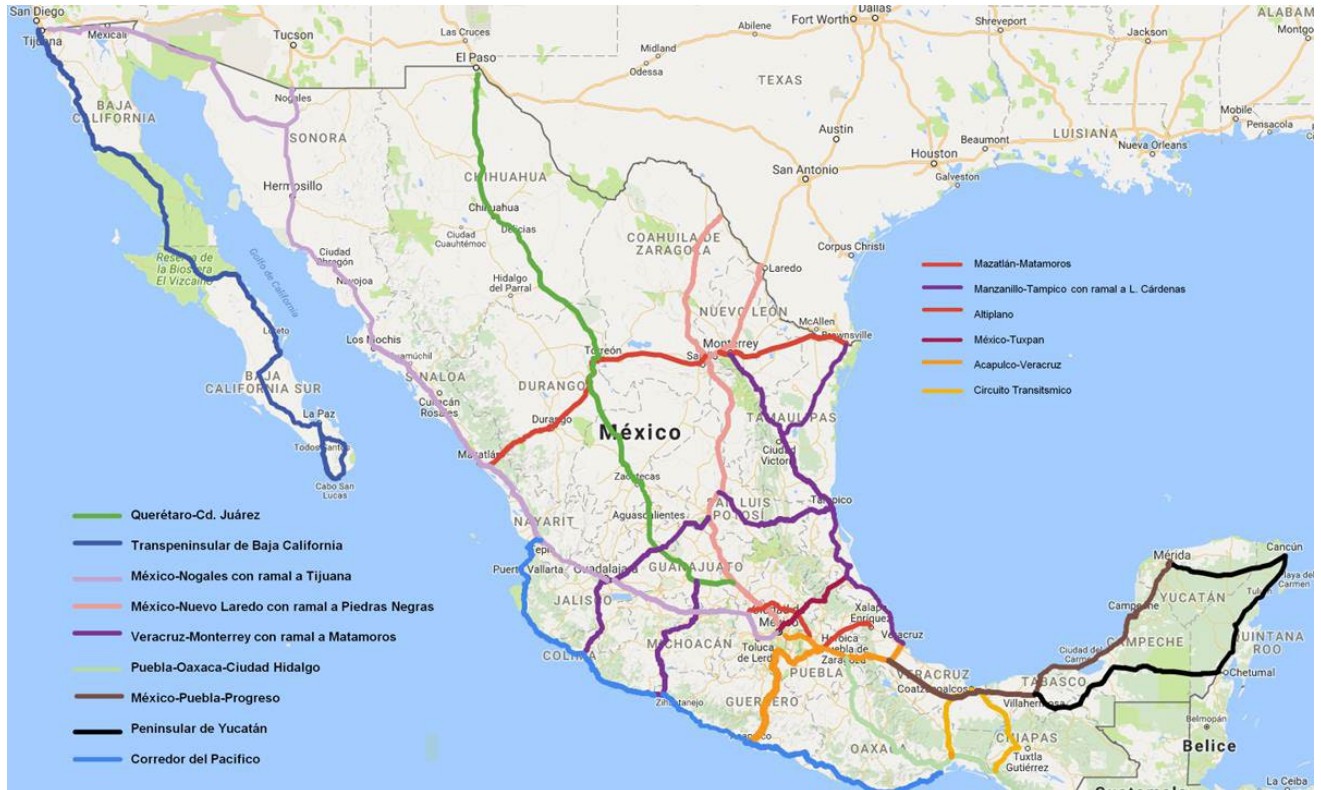

**Figure 2.** Transport road corridors in Mexico. Source: Own elaboration using Google Maps.

### 3.2. Traffic Data

The traffic data required are the average annual daily traffic (AADT) of each route in the different sections along its entire length. In Mexico, the SICT has installed and operates a system of traffic measuring, which keeps macroscopic traffic variables updated annually, such as volumes and speed information. Traffic data also allows vehicle classification (Table 1), according to the percentages extracted from the highway traffic data of the SICT [42].

**Table 1.** Vehicle Classification.

| FHWA Classification [43] [a] | Type of Vehicle | SICT Classification [42] [b] |
|---|---|---|
| Class 2 and 3 | Light Vehicle | A |
| Class 4 | Bus | B |
| Class 5 | Medium truck (2 axes) | C2 |
| Class 6 | Heavy truck (3 axes) | C3 |
| Class 9 | Articulated truck (5 axes) | T3-S2 |
| Class 10 | Articulated truck (6 axes) | T3-S3 |
| Class 13 | Double articulated truck (9 axes) | T3-S2-R4 |

Source: [a] Data from FHWA (2014) [43] and [b] from the SICT (2020) [42].

The vehicle fleet data provide detailed information for calculating and estimating speeds, operating costs, travel time, and other aspects. It is necessary to also identify the number of passengers, number of wheels and axles, annual kilometers, average vehicle life, working hours and weight, and other data.

The characterization was carried out according to the studies of the Origin and Destination Survey (ODs) that apply the SICT to the country's roads, which also includes the weights and dimensions of vehicles and the age of the vehicles traveling on the roads. Since the stations for the implementation of the surveys are located at different points every year, it is not possible to apply for a single study. Therefore, the present work decided to use the last results of the monitoring station located on each transport corridor [44]. A summary of the studies to determine the age of the vehicle fleet is shown in Table 2.

**Table 2.** Average age of the vehicle fleet.

| ID Station | Name | Road Section | Year | Transport Corridor | Average Age/Type of Vehicle | | | | | |
|---|---|---|---|---|---|---|---|---|---|---|
| | | | | | B | C2 | C3 | T3S2 | T3S3 | T3S2R4 |
| 274 | El Cerrito | Querétaro-San Luis Potosí | 2011 | 1 | 6.5 | 8.1 | 13.1 | 8.1 | 9.2 | 6.1 |
| 257 | Samalayuca III | El Sueco-Ciudad Juárez | 2010 | 2 | 8.1 | 7.6 | 13.1 | 10 | 11.9 | 7.6 |
| 249 | Las Viguitas | Hermosillo-Nogales | 2010 | 3* | 6.3 | 8.1 | 14.7 | 13.1 | 13.4 | 8.7 |
| 259 | El Valiente | Ciudad Obregón-Hermosillo | 2010 | 3* | 7 | 8.4 | 12.7 | 10.2 | 12.2 | 6.9 |
| 229 | PC Cuyutlán | Armería-Manzanillo | 2009 | 4 | 6.9 | 7.7 | 9.3 | 7.3 | 6.4 | 4.8 |
| 154 | Canal SARH | Córdoba-Veracruz (cuota) | 2002 | 5 | 7.3 | 10.1 | 8.6 | 8.6 | 4.9 | 5.6 |
| 279 | Vega de Alatorre | Poza Rica-Veracruz | 2011 | 6 | 6.8 | 9.4 | 16.4 | 9.3 | 9.3 | 3.6 |
| 271 | Los Tamarindos | Coatzacoalcos-Salina Cruz | 2011 | 7 | 10.7 | 11.1 | 12.7 | 12.1 | 11 | 6.9 |
| 264 | Tabasquillo | Villa Hermosa-Cd. Carmen | 2010 | 8 | 6.4 | 7.5 | 9.7 | 8.3 | 8.9 | 6 |
| 283 | Bacalar | Reforma Agraria-Puerto Juárez | 2011 | 9 | 9 | 9 | 13.2 | 10.3 | 14.4 | 6.5 |
| 228 | San Felipe | Playa Azul-Manzanillo | 2009 | 10 | 10 | 10.3 | 18.4 | 11 | 12.4 | 14.3 |
| 205 | La Rosa | Saltillo-Torreón | 2006 | 11 | 8.3 | 8.3 | 12.1 | 10.1 | 10.5 | 8.7 |
| 245 | Popotla | Tijuana-Ensenada | 2010 | 12 | 14.5 | 12.5 | 13 | 16.3 | 15 | 8.1 |
| 237 | Cuapiaxtla | Amozoc-Perote | 2009 | 13 | 8.5 | 10.5 | 15.6 | 8.3 | 10.8 | 5.6 |
| 292 | Palo Blanco | Cuernavaca-Acapulco | 2012 | 14 | 6.5 | 8.7 | 12.9 | 9.2 | 9.1 | 6.2 |
| 292 | Palo Blanco | Cuernavaca-Acapulco | 2011 | 15 | 6.5 | 8.7 | 12.9 | 9.2 | 9.1 | 6.2 |
| 268 | Zacatal | Coatzacoalcos-Salina Cruz | 2011 | 16 | 10.6 | 10.09 | 17.7 | 13.4 | 15 | 7.1 |

Source: Own elaboration based on Mexican Institute of Transportation (2002, 2009, 2010, 2011) [44].

### 3.3. Geography and Environmental Characterization

In order to use the model, climatological data are acquired, such as precipitation, humidity, temperature, and their average ranges of variation. Köppen climate classification adapted was used to characterize the environmental regions in the country [45]. This scale consists of a worldwide climatic classification that identifies each type of climate with a series of letters, indicating the behavior of temperature and rainfall that characterize this type of climate. Additionally, the transport corridors of the study identified eight climate types (Table 3).

**Table 3.** Climate classification in México.

| No. | Climate Zone | Description |
|:---:|:---:|:---:|
| 1 | Af | Tropical with year-round rainfall |
| 2 | Am | Tropical with monsoon rains (short dry) |
| 3 | Aw | Tropical with summer rains (winter dry) |
| 4 | Bs | Semi-arid (steppe climate) |
| 5 | Bw | Arid (desert climate) |
| 6 | Cf | Temperate with year-round rainfall |
| 7 | Cw | Temperate with summer rain (dry winter) |
| 8 | Cs | Temperate with winter rains (dry summers) |

Source: Own elaboration based on García (1998) [45].

### 3.4. Design Infrastructure

The road geometry is related to the angular changes of the vertical and horizontal alignment (curves of the road) and is closely related to the topography. This information was obtained from the road design databases provided by the Secretariat of Infrastructure, Communications and Transportation [46].

The required data on the vertical geometry for the model are elevation (in meters) above mean sea level, for each section, as well as the corresponding vertical slopes in this case per kilometer, which are entered as a percentage, in proportion with the angle of the slope and to the horizontal, the starting point to the end point of the section.

The status indicator that assesses the roads, in terms of the quality of the surface layer for the bearing of vehicles, is the International Roughness Index (IRI). The IRI was developed by the World Bank in the 1980s [47]. The IRI is used to define a characteristic of the longitudinal profile of a traveled wheel track and constitutes a standardized roughness measurement. The commonly recommended units are meters per kilometer (m/km) or millimeters per meter (mm/m). The correlation between the IRI, speed, and vehicle operating costs is one of the most important premises on which the analysis methodology is based, using HDM-4 [36]; parameters that are directly related to fuel consumption, by which the model predicts an approximate of the amount of emissions generated by vehicles.

Surface roughness information is obtained through field inspections conducted with different pieces of equipment across the federal road network of the country (freeways and tollways). The IRIs used for each road section and each corridor are averaged to obtain homogeneous sections, according to the sections of road that were defined mainly by the AADT.

## 4. Methodological Procedure to Estimate Emissions in Road Transport Corridors

The methodology uses the Social and Environmental Effects (SEA) sub-model of the Highway Development and Management System, known as HDM-4 [36]. The HDM-4 is useful because this tool was specifically designed for roads, and the main objective of this research work is to estimate emissions from road transport in major freight corridors of the country.

The sub-model SEA of HDM-4 calculates at a macro level, the quantity of emissions in the form of chemicals and noise generated by the operation of transport on a road network or road segment, and the energy balance of the life cycle of conservation strategies (see Figure 3).

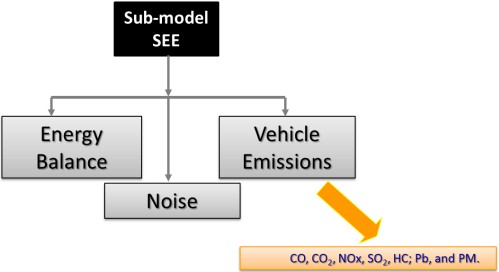

**Figure 3.** Structure of sub-model of Environmental Effects HDM-4. Source: Own elaboration based on World Road Association (PIARC) [36].

The HDM-4 emission model is based on the proposal by Hammarstrom [48] and predicts emissions from vehicle exhaust in terms of fuel consumption and average speed. Relationships and model coefficients were adjusted in terms of grams per kilometer [49].

Vehicle emission inventory based on a bottom-up approach requires the following information: emission factors according to the number of vehicles, vehicle age, and type of vehicle; the distances traveled by each type of vehicle; environmental data to characterize the study area, such as altitude and temperature; the quality state of the road network and comprising elements such as the pavement surface, operating speeds [32].

The proposed methodology for estimating emissions for transport operations on roads consists of three stages, as shown in Figure 4. The main effort is focused on the first stage, which is necessary to collect traffic data from the transport corridors, as well as obtaining information about the road geometry (horizontal and vertical alignment), characterization of the vehicle fleet (classification, age, etc.), environmental data (altitude, climate zone, etc.), where each road section is located, and the quality of pavement surface thought the IRI (see Section 3 "Methodology and Data Collection"). Stage 2 consists of the data capture process within the HDM-4 to perform all analyses. In Stage 3, the output of the analysis derived from the emissions model is obtained. The following pollutant emissions are estimated: hydrocarbons, nitrogen oxides, sulfur oxides, carbon monoxide, carbon dioxide, and particulate matter [50].

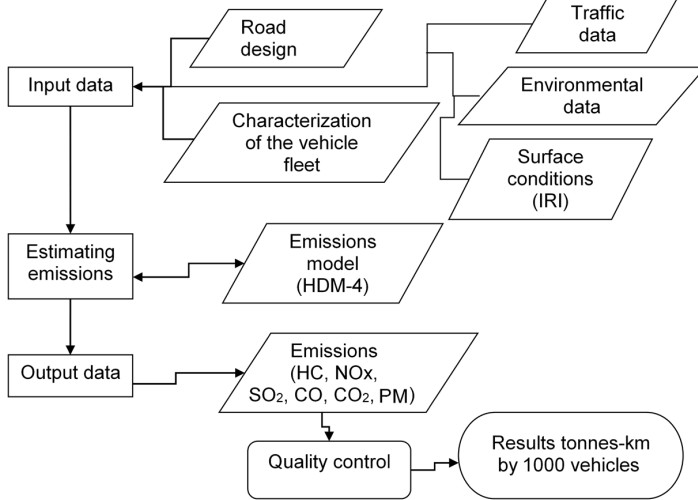

**Figure 4.** Methodology for estimating emissions of motor vehicles on road corridors. Source: Own elaboration. Notes: hydrocarbons (HC), nitrogen oxides (NOx), sulfur oxides ($SO_2$), carbon monoxide (CO), carbon dioxide ($CO_2$) and particulate matter (PM). The results are expressed in tons per kilometer per thousand vehicles.

## 5. Inventory of Emissions in Road Transport Corridors

In the emission inventory, the daily number of gases emitted into the atmosphere by the operation of transport vehicles through the transport corridors studied has been estimated by 2019. For this analysis, the most complete AADT road data were used.

Each transport corridor was divided into routes (homogeneous road sections and sections according to the surface condition of the road) to simplify the analyses and interpretation of results.

As the first part of an emission inventory, an analysis was developed of all transport corridors, which includes six pollutants quantification of all vehicles passing daily on different roads that integrate the corridors analyzed (hydrocarbons, nitrogen oxides, sulfur oxides, carbon monoxide, carbon dioxide, and suspended particulates).

From the result of the total emissions emitted, it was necessary to extract only those emissions that were generated by vehicles for freight transport. Vehicles used for goods and that circulate mostly on roads in Mexico are shown in Table 1. These vehicles are representative of the emission inventory for freight or goods vehicles. The total $CO_2$ emissions of each transport corridor for freight transport are shown in Table 4.

**Table 4.** Tonnes of $CO_2$ emitted daily by the corridor.

| Number | ID Mexican Corridor | Length of Corridor (Kilometers) | CO$_2$ Emissions (Tonnes) |
|---|---|---|---|
| 1 | México-Nuevo Laredo | 1155.61 | 17,897.68 |
| 1.1 | • Ramal a Piedras negras | 316.41 | 961.82 |
| 2 | México-Cd Juárez | 2003.14 | 13,687.80 |
| 3 | México-Nogales | 2437.46 | 10,850.56 |
| 3.1 | • Ramal a Tijuana | 761.13 | 1770.27 |
| 4 | Manzanillo–Tampico | 1364.27 | 6804.20 |
| 4.1 | • Ramal a Lázaro Cárdenas | 609.69 | 2733.26 |
| 5 | México-Veracruz | 389.16 | 6224.61 |
| 6 | Veracruz-Monterrey con ramal a Matamoros | 1291.32 | 3920.84 |
| 7 | Puebla-Oaxaca-Cd. Hidalgo | 1095.14 | 2289.78 |
| 8 | México-Puebla-Progreso | 1302.35 | 8482.65 |
| 9 | Peninsular de Yucatán | 1203.24 | 2068.97 |
| 10 | Corredor del Pacífico | 2045.48 | 1668.17 |
| 11 | Mazatlán-Matamoros | 1348.29 | 5376.85 |
| 12 | Transpeninsular de Baja California | 1878.61 | 1727.22 |
| 13 | Altiplano | 587.15 | 4524.24 |
| 14 | Acapulco-Tuxpan | 1022.84 | 3345.52 |
| 15 | Acapulco-Veracruz | 766.81 | 3715.08 |
| 16 | Circuito Transístmico | 799.57 | 1467.88 |

Source: Own elaboration.

The indicators of emissions include only $CO_2$ emissions generated by freight vehicles, in order to model an environmental indicator in terms of the length of each corridor.

## 6. Environmental Indicators for Road Transport Corridors

The purpose of an environmental indicator is to identify the size of the effect according to both the descriptive effect to characterize and the estimation of the magnitude in absolute



terms. To understand an indicator, this is something that provides a clue to a matter of larger significance or makes perceptible a trend or phenomenon [51].

There are different types of indicators for this study. The environmental impact indicators were developed according to the PSR model (Pressure-State-Response). The pressure indicator describes developments in the release of substances (emissions), physical and biological agents, the use of resources, and the use of land [52,53].

Then, in this study, the indicator to be used will be one of the pressures, which expresses the amount of emissions emitted into the atmosphere by the operation of freight vehicles in the transportation corridors.

The emission indicators were based only on $CO_2$ emissions generated by freight transport in order to model an environmental indicator considering the length of each corridor. The total emissions of each corridor were determined. Those from freight vehicles were extracted, and the effective lengths of each corridor were measured. The emissions are measured daily, as the AADT is used, but it is also possible to obtain the total annual emissions, multiplying daily emissions by 365. Table 5 shows the parameters for the environmental indicator used.

**Table 5.** Indicator 1, $CO_2$ emissions per kilometer for freight vehicles.

| Characteristic | Description |
|---|---|
| Indicator: | $CO_2$ emissions per kilometer |
| Period: | Annual/Daily |
| Unit: | Tonnes/Kilometer |
| What it measure? | The quantity of $CO_2$ emissions per kilometer of freight vehicles. |
| Type of Indicator: | Pressure |

Indicator 1 for each corridor is expressed in the Equation (1):

$$E_{CO_2} = \frac{\sum CO_2 \, emissions \, of \, each \, corridor}{effective \, length \, of \, each \, corridor} \tag{1}$$

The amount of $CO_2$ generated is expressed in tonnes, the effective length of the corridor is expressed in kilometers, and the direct result ($E_{CO_2}$) is expressed in tonnes per day of $CO_2$ per kilometer. Table 6 shows the results of indicator 1 for each road transport corridor.

**Table 6.** Indicator 1: Tonnes of $CO_2$ per kilometer by corridor.

| Number | ID Mexican Corridor | Length of Corridor (Kilometers) | $CO_2$ Emissions (Tonnes) | Indicator 1 "Tonnes of $CO_2$ per Kilometer" |
|---|---|---|---|---|
| 1 | México City-Nuevo Laredo | 1155.61 | 17,897.68 | 15.49 |
| 1.1 | • Ramal a Piedras negras | 316.41 | 961.82 | 3.04 |
| 2 | México City-Cd Juárez | 2003.14 | 13,687.80 | 6.83 |
| 3 | México City-Nogales | 2437.46 | 10,850.56 | 4.45 |
| 3.1 | • Ramal a Tijuana | 761.13 | 1770.27 | 2.33 |
| 4 | Manzanillo–Tampico | 1364.27 | 6804.20 | 4.99 |
| 4.1 | • Ramal a Lázaro Cárdenas | 609.69 | 2733.26 | 4.48 |
| 5 | México City-Veracruz | 389.16 | 6224.61 | 15.99 |
| 6 | Veracruz-Monterrey con ramal a Matamoros | 1291.32 | 3920.84 | 3.04 |
| 7 | Puebla-Oaxaca-Cd. Hidalgo | 1095.14 | 2289.78 | 2.09 |
| 8 | México City-Puebla-Progreso | 1302.35 | 8482.65 | 6.51 |
| 9 | Peninsular de Yucatán | 1203.24 | 2068.97 | 1.72 |
| 10 | Corredor del Pacífico | 2045.48 | 1668.17 | 0.82 |

**Table 6.** *Cont.*

| Number | ID Mexican Corridor | Length of Corridor (Kilometers) | CO$_2$ Emissions (Tonnes) | Indicator 1 "Tonnes of CO$_2$ per Kilometer" |
|---|---|---|---|---|
| 11 | Mazatlán-Matamoros | 1348.29 | 5376.85 | 3.99 |
| 12 | Transpeninsular de Baja California | 1878.61 | 1727.22 | 0.92 |
| 13 | Altiplano | 587.15 | 4524.24 | 7.71 |
| 14 | Acapulco-Tuxpan | 1022.84 | 3345.52 | 3.27 |
| 15 | Acapulco-Veracruz | 766.81 | 3715.08 | 4.84 |
| 16 | Circuito Transístmico | 799.57 | 1467.88 | 1.84 |
| Average | | | | 3.51 |

According to the results obtained, the Mexico City-Nuevo Laredo corridor is the main generator of CO$_2$ emissions per kilometer, mainly because it is the corridor with the most cargo traffic in the country connecting the USA and Canada.

To determine this indicator to estimate CO$_2$ emissions generated by freight vehicles according to the number of tonnes transported per kilometer, additional information is required from the Origin-Destination Survey, effective cargo moving on the road corridor.

Emissions per kilometer were determined with indicator 1. The total weight is determined by information on the gross weight of the loaded and empty vehicles, the percentage of vehicles loaded and empty, and the weight of the average load of vehicles loaded, using Equation (2).

$$Weight\ of\ load = \frac{Weight\ of\ the\ average\ load\ *\ Percentage\ of\ vehicles\ loaded}{100} \tag{2}$$

Once the weight of the load has been determined, daily tonnes transported can be estimated by Equation (3).

$$Weight\ daily\ tons = AADT_{Heavy\ vehicles} * Weight_{load} \tag{3}$$

Finally, indicator 2 ($T_{CO_2*Ton-km}$) is expressed as indicate in Equation (4)

$$T_{CO2*Ton-km} = \frac{Emissions\ tons\ of\ CO_2 * 1 * 10^6}{Length\ of\ corridors\ *\ Weight\ daily\ tons} \tag{4}$$

Table 7 details the characteristics of indicator 2.

**Table 7.** Indicator 2: Tonnes of CO$_2$ per million tonne-kilometers.

| Characteristic | Description |
|---|---|
| Indicator: | Tonnes of CO$_2$ per million tonne-kilometer for freight transport |
| Period: | Daily |
| Unit: | Tonnes |
| What it measure? | The quantity of CO$_2$ emissions on average by freight vehicles on federal highways of Mexico per million tonnes transported per kilometer |
| Type of Indicator: | Pressure |
| Geographic coverage: | National |

The results of the calculation of indicator 2 "Tonnes of CO$_2$ per million tonne-kilometer" for each road transport corridor analyzed are shown in Table 8. For this analysis, the length of corridors considers both directions, which may vary when the road has two separate sections.

**Table 8.** Indicator 2: Tonnes of $CO_2$ per million Tonnes-kilometer.

| Number | ID Mexican Corridor | Length of Corridor Analyzed (Kilometers) | $CO_2$ Emissions (Tonnes) | Weight Daily (Tonnes) | Indicator 2 "Tonnes of $CO_2$ per million tonne-kilometers" |
|---|---|---|---|---|---|
| 1 | México City-Nuevo Laredo | 2279.02 | 17,897.68 | 64,993 | 120.83 |
| 1.1 | • Ramal a Piedras negras | 543.12 | 961.82 | 19,320 | 91.66 |
| 2 | México City-Cd Juárez | 3656.52 | 13,687.80 | 26,912 | 139.10 |
| 3 | México City-Nogales | 4484.07 | 10,850.56 | 20,037 | 120.77 |
| 3.1 | • Ramal a Tijuana | 1057.24 | 1770.27 | 16,001 | 104.65 |
| 4 | Manzanillo–Tampico | 2050.25 | 6804.20 | 61,085 | 54.33 |
| 4.1 | • Ramal a Lázaro Cárdenas | 836.29 | 2733.26 | 31,478 | 103.83 |
| 5 | México City-Veracruz | 778.32 | 6224.61 | 47,055 | 169.96 |
| 6 | Veracruz-Monterrey con ramal a Matamoros | 1758.48 | 3920.84 | 22,400 | 99.54 |
| 7 | Puebla-Oaxaca-Cd. Hidalgo | 1581.65 | 2289.78 | 7231 | 200.21 |
| 8 | México City-Puebla-Progreso | 2408.43 | 8482.65 | 18,831 | 187.04 |
| 9 | Peninsular de Yucatán | 1757.18 | 2068.97 | 9966 | 118.15 |
| 10 | Corredor del Pacífico | 2387.46 | 1668.17 | 5213 | 134.03 |
| 11 | Mazatlán-Matamoros | 2423.63 | 5376.85 | 7205 | 307.91 |
| 12 | Transpeninsular de Baja California | 2296.0 | 1727.22 | 7571 | 99.36 |
| 13 | Altiplano | 1109.2 | 4524.24 | 15,604 | 261.40 |
| 14 | Acapulco-Tuxpan | 1833.13 | 3345.52 | 15,343 | 118.95 |
| 15 | Acapulco-Veracruz | 1453.51 | 3715.08 | 15,419 | 165.77 |
| 16 | Circuito Transístmico | 953.25 | 1467.88 | 11,218 | 137.27 |
| Average | | | | | 143.93 |

## 7. Modeling Baseline $CO_2$ Emissions

The modeling of the baseline through Geographic Information Systems (GIS) was based on the main corridors. These maps allow visualization of where the highest concentration of $CO_2$ emissions per kilometer of freight transported exists. Figure 5 shows the model based on Google Maps, such as the annual emissions per kilometer (environmental indicator 1).

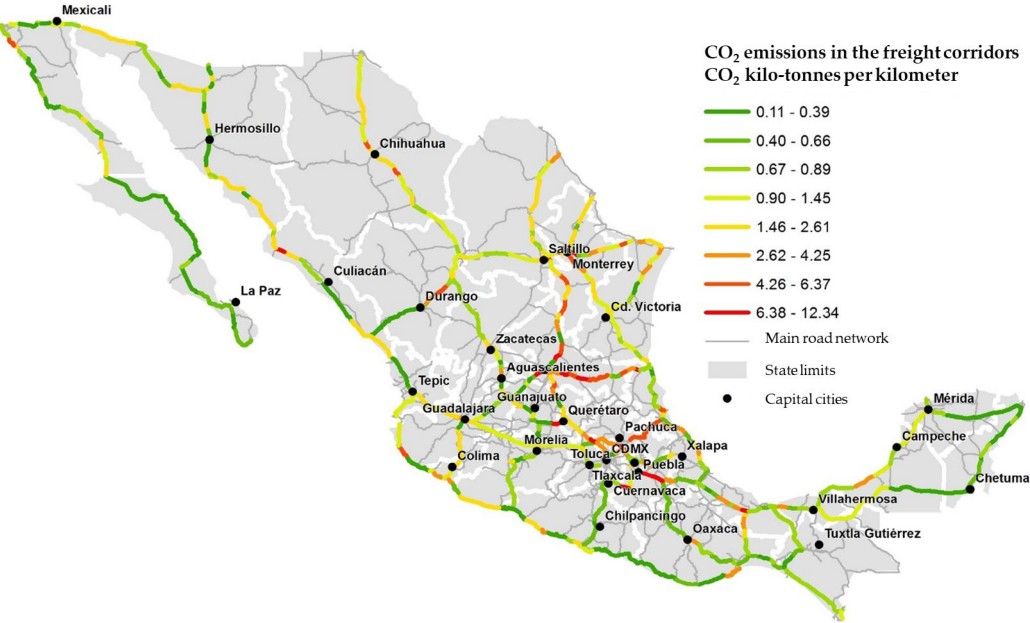

**Figure 5.** Modeling of $CO_2$ emissions in the freight corridors. Source: Own elaboration using Google Maps.

Based on the results obtained in the indicators and using the historical growth of heavy vehicles in the corridors, a trend line for freight transportation in Mexico can be constructed [54]. However, the prediction of a trend may not be fully accurate depending on the model used for forecasting [55].

To estimate the growth rate in the freight transportation sector, the growth of the vehicle fleet and cargo transported (including the contraction due to the COVID-19 pandemic) was used, and from the period 2013 to 2020, we obtained a growth of 3.53%, which can be applied to forecasting the growth of emissions over the 2020 baseline estimate of 36.32 Megatonnes of $CO_2$ (99,517.4 tonnes daily). Figure 6 shows the growth of baseline $CO_2$ emissions in freight transportation in the main road corridors in Mexico.

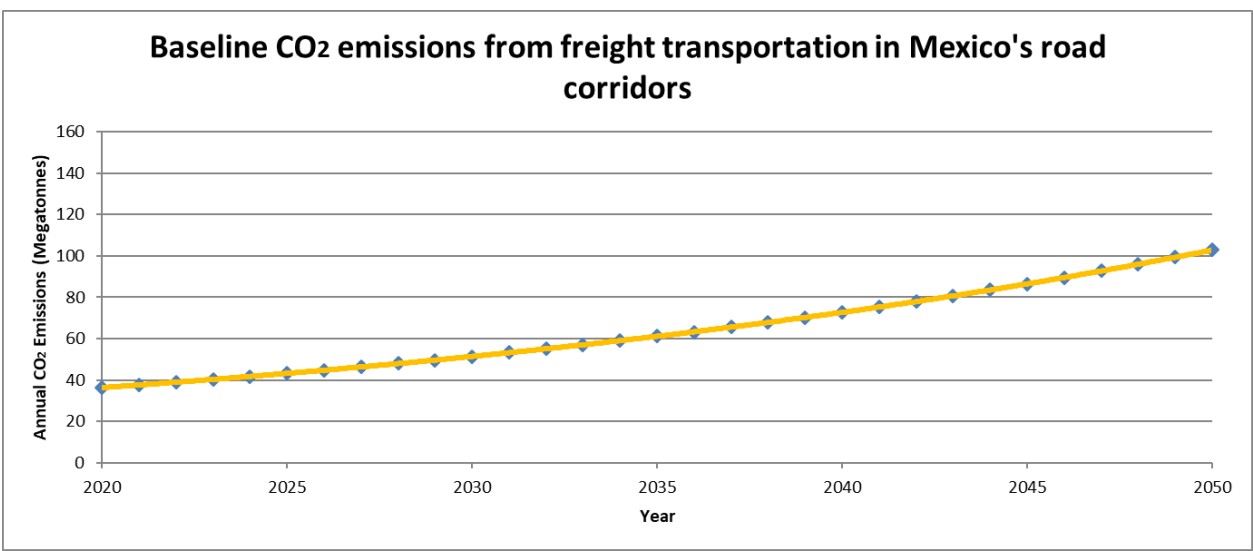

**Figure 6.** Baseline $CO_2$ emissions from freight transportation in Mexico's road corridors. Source: Own elaboration.

## 8. Conclusions

The proposed methodology has the advantage that it considers in a very detailed way the geometric characteristics of the road, which have an important influence on fuel consumption and emissions generation.

The HDM-4 model was very useful for estimating emissions, as it is developed specifically for roads, and the information required allows for a bottom-up analysis according to the methodologies that are used worldwide to estimate emissions at a higher level of reliability.

The results of $CO_2$ emissions allow the ability to propose the construction of two environmental indicators of pressure on transport corridors: (i) Tonnes of $CO_2$ per kilometer by corridor, and (ii) Tonnes of $CO_2$ per million tonnes-kilometer. The first indicator reflects the number of daily emissions in tonnes emitted into the atmosphere per kilometer of freight in the corridors analyzed, whilst the second represents $CO_2$ emissions per tonne per kilometer of goods transported in transport corridors analyzed.

These indicators make it possible to establish a diagnosis of freight transportation emissions in Mexico and build a baseline, through which mitigation strategies to reduce GHG emissions can be proposed. This allows the generation of public and private policies to respond to climate change.

The study shows that for Mexico, there is a growth trend in $CO_2$ emissions produced by road transportation. The results will establish appropriate scenarios to understand the behavior of emissions from freight transport in Mexico and its impact on climate change.

Although the methodology requires multiple data for its analysis, these data are easy to obtain by road organizations; hence, the replication of this study will be very easy for other countries.

The following phases of this research will be used to develop guidelines for decision-makers in the transport sector to help them choose between well-documented strategies to mitigate GHG emissions in the road sector as action against climate change and for optimal use of monetary resources.

With this information, future scenarios can be constructed based on changes in model variables, such as an increase in the AADT or road deterioration through the IRI.

To mitigate climate change, it is important to implement strategies to decarbonize transportation. Using e-fuels, hybrid electric vehicles and electrification (vehicles and infrastructure) could have potential benefits to reduce emissions in road transport corridors in Mexico. Moreover, for sustainability, it is important to consider environmental considerations in freight transportation.

**Author Contributions:** Methodology, J.F.M.-S., E.M.A.-G. and R.S.-E.; Investigation, J.F.M.-S., W.M.-M. and H.L.C.-G.; Resources, H.D.-A.; Writing—original draft, J.F.M.-S. and E.M.A.-G.; Writing—review & editing, S.A.O.-B.; Supervision, E.M.A.-G. All authors have read and agreed to the published version of the manuscript.

**Funding:** This research was financed by the Mexican Institute of Transportation and the Michoacan University of San Nicolas of Hidalgo, through research funds, including the support of the National Council for the Humanities, Science and Technology.

**Institutional Review Board Statement:** Not applicable.

**Informed Consent Statement:** Not applicable.

**Data Availability Statement:** Data is contained within the article.

**Conflicts of Interest:** The authors declare no conflict of interest.

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
