# Peer review of "Measurement of CO2 Emissions by the Operation of Freight Transport in Mexican Road Corridors"

_applsci, doi:10.3390/app132011391_

Round 1

Reviewer 1 Report

This paper describes a methodology for the estimation of freight vehicles CO2 emissions in main transport corridors.

The entire approach is well described including the data collection and the approach used for emission estimation. Only Section 7, which is focused on modeling of baseline CO2 emissions should be expanded to fully explain the approach used (not only referring to [55] and [56].

On the other hand, the structure and headings are the major issues of the paper. With a few exceptions in Section 3 (where there are unnumberred subsections), there are no subsections. Additionally, the font family and size are the same for level 1 headings, figures and table captions.  Also, there is quite large left indentation of the text, but not of te figures and tables. These problems complicate the readability of the paper. To correct this, the entire structure of the sections should be changed to contain numberred subsections (with the exception of Introduction and Conclusion, which usually do not contain subsections).

The figures are adequate and legible.

The references are relevant and relatively up-to-date.

The English is quite good, the number of typos and errors is low.

Reviewer 2 Report

The authors have developed a model to predict CO2 emission from transportation of freight in Mexico.  The model includes factors such as the number of vehicles, the percentage of maximum load carried, the age of the fleet, the road surfaces and geometry, and the climatic conditions.  Their model would be straightforward to implement for other regions and therefore has great value.  The manuscript is very well written and the model clearly described. 

Reviewer 3 Report

After getting familiar with the reading-matter of reviewed article, I state the following:
1. The article presents a methodology for estimating CO2 emissions for freight vehicles on road transport corridors in Mexico. CO2 emissions are characterized by two indicators: CO2 emissions per kilometr and CO2 emissions per tonne transported.
2. The article is well written, both in terms and editing.
3. In tables: 4 (columns 3 and 4), 6 (columns 3 and 4) and 8 (columns 3, 4 and 5) - numerical values, starting with four digits, should be written with a comma every three digits.
4. The article may be published in Applied Sciences after taking into account the comments.

Reviewer 4 Report

This paper represents a work concerning a methodology for the estimation of emissions for freight vehicles on road transportation corridors.

I consider the work presented sufficiently structured and I propose only a minor review, hoping that the suggestions I propose can improve the quality of the document.

In the attached PDF file You will find some suggestions and observations, with reference to the lines of the text or to the figures, equations, or sections of the paper.
